# Enhanced exercise and regenerative capacity in a mouse model that violates size constraints of oxidative muscle fibres

Saleh Omairi[1†], Antonios Matsakas[2†], Hans Degens[3,4], Oliver Kretz[5,6], Kenth-Arne Hansson[7], Andreas Våvang Solbrå[7,8], Jo C Bruusgaard[7,9], Barbara Joch[10,11], Roberta Sartori[12], Natasa Giallourou[13], Robert Mitchell[1], Henry Collins-Hooper[1], Keith Foster[1], Arja Pasternack[14], Olli Ritvos[14], Marco Sandri[12], Vihang Narkar[15], Jonathan R Swann[13], Tobias B Huber[5,6,16,17], Ketan Patel[1,17]*

[1]School of Biological Sciences, University of Reading, Reading, United Kingdom; [2]Hull York Medical School, Hull, United Kingdom; [3]School of Healthcare Science, Manchester Metropolitan University, Manchester, United Kingdom; [4]Lithuanian Sports University, Kaunas, Lithuania; [5]Renal Division, University Medical Center Freiburg, Freiburg, Germany; [6]Faculty of Medicine, University of Freiburg, Freiburg, Germany; [7]Centre for Integrative Neuroplasticity, Department of Biosciences, University of Oslo, Oslo, Norway; [8]Department of Physics, University of Oslo, Oslo, Norway; [9]Department of Health Sciences, Kristiania University College, Oslo, Norway; [10]Department of Neuroanatomy, University of Freiburg, Freiburg, Germany; [11]Faculty of Medicine, University of Freiburg, Freiburg, Germany; [12]Venetian Institute of Molecular Medicine, University of Padua, Padua, Italy; [13]Department of Food and Nutritional Sciences, University of Reading, Reading, United Kingdom; [14]Department of Bacteriology and Immunology, Haartman Institute, University of Helsinki, Helsinki, Finland; [15]Institute of Molecular Medicine, University of Health Science Center, Houston, Texas; [16]BIOSS Center for Biological Signalling Studies, Albert-Ludwigs-University Freiburg, Houston, Texas; [17]FRIAS, Freiburg Institute for Advanced Studies and Center for Biological System Analysis ZBSA, Freiburg, Germany

*For correspondence: ketan.
patel@reading.ac.uk

†These authors contributed
equally to this work

Competing interests: The
authors declare that no
competing interests exist.

Reviewing editor: Giulio Cossu,
University of Manchester, United
Kingdom

**Abstract** A central tenet of skeletal muscle biology is the existence of an inverse relationship between the oxidative fibre capacity and its size. However, robustness of this relationship is unknown. We show that superimposition of Estrogen-related receptor gamma (*Errγ*) on the myostatin (Mtn) mouse null background ($Mtn^{-/-}/Errγ^{Tg/+}$) results in hypertrophic muscle with a high oxidative capacity thus violating the inverse relationship between fibre size and oxidative capacity. We also examined the canonical view that oxidative muscle phenotype positively correlate with Satellite cell number, the resident stem cells of skeletal muscle. Surprisingly, hypertrophic fibres from $Mtn^{-/-}/Errγ^{Tg/+}$ mouse showed satellite cell deficit which unexpectedly did not affect muscle regeneration. These observations 1) challenge the concept of a constraint between fibre size and oxidative capacity and 2) indicate the important role of the microcirculation in the regenerative capacity of a muscle even when satellite cell numbers are reduced.

## Introduction

John Eccles and colleagues first applied the concept of 'plasticity' to skeletal muscle to describe the effect of cross-innervation experiments in cats on the size and fibre characteristics of skeletal muscle (*Buller et al., 1960*). Many factors have since been shown to profoundly effect on skeletal muscle structure and function, including chronic electrical stimulation, exercise, diet and ageing (*Salmons and Vrbová, 1969*; *Hickson, 1980*; *Wade et al., 1990*; *Mitchell et al., 2012*).

In mammalian skeletal muscle, fibres are broadly characterized as slow or fast fibres, where slow fibres express the myosin heavy chain (MHC) isoform I, whereas fast fibres express MHC IIA, IIX and/ or IIB. Slow fibres generally have a smaller cross sectional area (CSA), contain more mitochondria which sustain a high oxidative capacity, and a denser microvascular network than fast fibres that rely predominantly on glycolysis for ATP production. Muscle fibres can change their phenotype, such as the expression of MHC, mitochondrial content and capillary supply in response to external stimuli (*Pette and Staron, 1997*, *2001*).

We are beginning to understand some of the cellular, biochemical and molecular processes that act to concord muscle structure and morphology to the functional demands placed on the muscle. For instance, it has been shown that the development of the slow muscle fibre phenotype is largely controlled by Protein Kinase C, Calcineurin/NFAT, AMP Activated Protein kinase (AMPK), peroxi-some proliferator-activated receptor gamma co-activator 1-alpha (PGC-1α) and Sex determining region Y-box 6 (Sox6) (*Gundersen, 2011*; *von Hofsten et al., 2008*). Recently, we have shown that the Estrogenrelated receptor gamma (Errγ) is robustly expressed in slow muscle and can promote the formation of oxidative fibres in a PGC-1α independent manner (*Narkar et al., 2011*). Fast, glyco-lytic muscle development on the other hand seems to involve the activation of the Akt signalling pathway through the transcriptional regulation by molecules including Baf60c (also called Smarcd3) and T-box 15 (Tbx15) (*Meng et al., 2013*, *2014*; *Lee et al., 2015*). Lifting the inhibition of Akt sig-nalling mediated by Myostatin is also a potent means of inducing the formation of glycolytic muscle fibres (*Trendelenburg et al., 2009*). Additionally, a recent study has shown that the DNA binding protein Nuclear Factor I X (Nfix) acts to inhibit the slow muscle phenotype (*Rossi et al., 2016*).

Myostatin (Mtn), a member of the Transforming Growth Factor Beta (TGF-β) family of secreted proteins, is highly expressed in skeletal muscle (*McPherron et al., 1997*). It is a potent inhibitor of skeletal muscle growth and its deletion results in a hypermuscular phenotype called 'Muscle Dou-bling' seen in mice, cattle and even humans (*McPherron et al., 1997*; *McPherron and Lee, 1997*; *Schuelke et al., 2004*). We and others have shown that the glycolytic muscles that develop in the absence of Mtn have a mitochondrial deficit and a low specific force (*Amthor et al., 2007*; *Mendias et al., 2006*).

A fundamental concept of skeletal muscle biology is the existence of the inverse relationship between the oxidative capacity of a fibre and its cross-sectional area (CSA) that applies to muscles as diverse as the limb, diaphragm and masseter muscle within an animal and even across species boundaries (*van Wessel et al., 2010*; *Degens, 2012*; *Van Der Laarse et al., 1997*). This relationship, in theory, ultimately imparts a constraint on the size that mitochondria-rich and therefore high $O_2$ -dependent oxidative fibres can attain before they become anoxic or adapt to a glycolytic phenotype less reliant on $O_2$ (*Desplanches et al., 1996*; *Deveci et al., 2001*). The metabolic properties of mus-cle are believed not only to control fibre size but also the number of satellite cells. A number of cor-relative studies have described the number of SC increases as a muscle becomes progressively oxidative (*Putman et al., 1999*; *Christov et al., 2007*).

Here we investigated whether this suggested constraint between fibre size and oxidative capacity can be broken and sought to develop large oxidative fibres without compromising function, such as fatigue resistance. To that end, we developed a novel mouse line by introducing an Errγ over-expression allele driven by a skeletal muscle fibre promoter (Human α -Skeletal Muscle Actin) (*Muscat and Kedes, 1987*) that enhances the oxidative capacity (*Narkar et al., 2011*) into a hyper-trophic $Mtn^{-/-}$ background. Based on the concept of a constraint between the CSA and oxidative capacity of a fibre we postulated three possible outcomes of the cross: (1) the Akt pathway that is de-repressed due to the absence of Mtn would prevail and lead to hypertrophic, but glycolytic fibres; (2) oxidative features would be imparted by the Errγ programme that would follow the inverse size relationship and lead to mitochondria-rich fibres which could be smaller than wild-type

(*Rangwala et al., 2010*); (3) the constraint is broken in this strain and results in the development of hypertrophic yet oxidative fibres.

The main observations of the study are firstly that the muscles of $Mtn^{-/-}/Err\gamma^{Tg/+}$ mice have large fibres with a larger than expected oxidative capacity, breaking the constraint of the inverse size-oxidative capacity relationship. This was attained through the activation of the Akt pathway, increased myoglobin gene expression, relocation of mitochondria to the sub sarcolemma and hyper-capillarisation of the muscle. We show that these modifications not only bring about normalization of many ultrastructural abnormalities in the hypertrophic muscles of $Mtn^{-/-}$ mice, but the $Mtn^{-/-}/Err\gamma^{Tg/+}$ mice even outperform wild type mice during an incremental exercise test. Secondly that the hypertrophic oxidative muscles from the $Mtn^{-/-}/Err\gamma^{Tg/+}$ mice do not follow the dogma regarding metabolism and satellite cells number. We actually show that the metabolic reprogramming in this study led to a decrease in satellite cell number. However, this deficit did not impact at all in terms of the muscle's ability to regenerate. We believe this highlights the importance of the microcirculation during regeneration and has major clinical implications.

## Results

### Body and skeletal muscle mass

Introduction of Errγ in a skeletal muscle-specific manner into the $Mtn^{-/-}$ background to generate double transgenic $Mtn^{-/-}/Err\gamma^{Tg/+}$ resulted in viable, fertile offspring that were born at the expected Mendelian ratios. Firstly, we found that the HSA promoter used induced robust over-expression of Errγ in the $Mtn^{-/-}$ background (*Figure 1A*). The body mass of WT, $Mtn^{-/-}$ and $Mtn^{-/-}/Err\gamma^{Tg/+}$ animals was similar at 12 weeks of age (*Figure 1B*). However, the EDL, gastrocnemius, soleus and TA muscles were in both $Mtn^{-/-}$ and $Mtn^{-/-}/Err\gamma^{Tg/+}$ approximately 43%, 44%, 47% and 70% larger than their WT counterpart, respectively (*Figure 1C–F*). Importantly, there was no significant difference in mass for any of the muscles from $Mtn^{-/-}$ and $Mtn^{-/-}/Err\gamma^{Tg/+}$ mice (*Figure 1C–F*).

### Exercise capacity

Using the running to exhaustion protocol on a treadmill, we found that $Mtn^{-/-}$ mice performed worse than WT. However the $Mtn^{-/-}/Err\gamma^{Tg/+}$ ran for approximately 80% longer than the $Mtn^{-/-}$ and 25% longer than the WT mice (*Figure 1G*).

### Force generating capacity

We found that the maximal isometric tetanic force generated by the EDL of $Mtn^{-/-}$ was not significantly different from that of the WT mice, despite the larger muscle mass (*Figure 1H*). The tetanic force generated by $Mtn^{-/-}/Err\gamma^{Tg/+}$ EDL was, however, greater than that of the EDL from both WT and $Mtn^{-/-}$ mice. We next calculated the Specific Force ($sP_o$), the tetanic force per muscle mass. The $sP_o$ of the EDL of $Mtn^{-/-}$ mice was lower that of the other groups, with that of the $Mtn^{-/-}/Err\gamma^{Tg/+}$ mice being significantly greater than $Mtn^{-/-}$ mice, but not normalized to WT levels (*Figure 1I*). We also examined the force generating capacity of the soleus. The tetanic force of $Mtn^{-/-}$ soleus muscle was significantly lower than those of WT. There was no difference in this parameter between the soleus muscles of WT and $Mtn^{-/-}/Err\gamma^{Tg/+}$ (*Figure 1—figure supplement 1A*). The specific force of the soleus showed the same overall profile as that of the EDL but did not reach statistical significance, possibly due to low sample size (*Figure 1—figure supplement 1A*).

### Muscle fibre number, area and MHC profile

The increased muscle mass in $Mtn^{-/-}$ mice are due to both hypertrophy and hyperplasia. We found that the introduction of Errγ into $Mtn^{-/-}$ did not significantly change the number of fibres normally seen in $Mtn^{-/-}$ EDL (*Figure 2A–B*) or soleus muscles (*Figure 2—figure supplement 1A–B*) both of which were greater than in WT. The fibre sizes were equivalent in the EDL of $Mtn^{-/-}$ and $Mtn^{-/-}/Err\gamma^{Tg/+}$ mice. Of particular note was that the MHCIIB fibres in the EDL were approximately 270% larger in both $Mtn^{-/-}$ and $Mtn^{-/-}/Err\gamma^{Tg/+}$ compared to WT (*Figure 2B*). The other notable result was the smaller size of MHCIIA fibres in $Mtn^{-/-}/Err\gamma^{Tg/+}$ than $Mtn^{-/-}$, but they were still larger than those in the WT (*Figure 2B*).

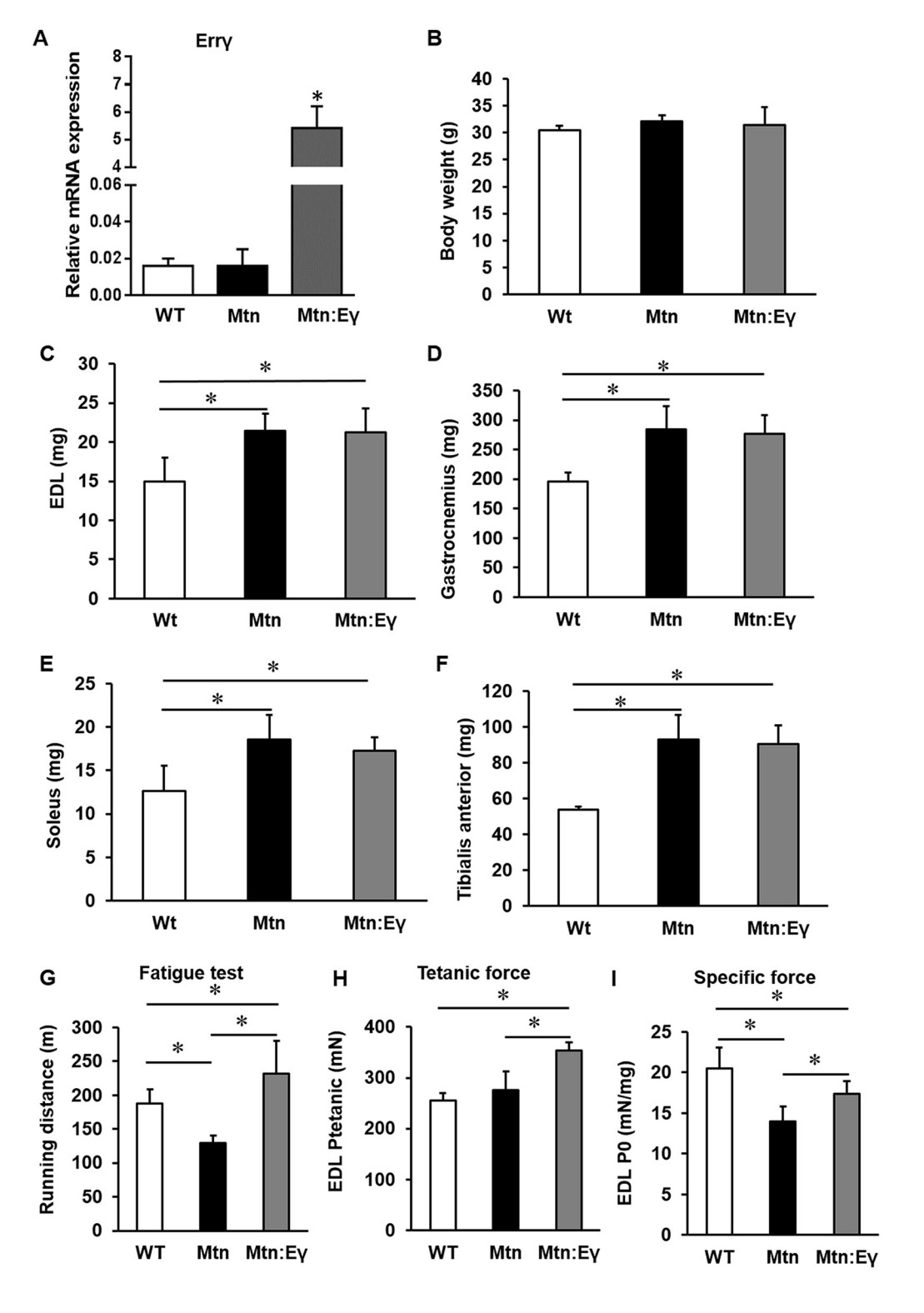

**Figure 1.** Concomitant skeletal muscle hypertrophy and tissue specific expression of ERRγ and resultant fatigue resistant characteristics. (**A**) *ERRγ* mRNA levels. (**B**) Body and (**C–F**) skeletal muscle mass in wild type (*Wt*), myostatin null (*Mtn*) and ERRγ transgenic mice on the myostatin null background (*Mtn:Eγ*). (**G**) Exercise tolerance test on a mouse treadmill. (**H–I**) Contractile properties of the EDL muscle. Specific force denotes tetanic force normalized to wet muscle mass. N = 5 male twelve-week old mice per group; One-way ANOVA followed by Bonferroni's multiple comparison tests, *p<0.05.

*Figure 1 continued on next page*

*Figure 1 continued*

The following figure supplement is available for figure 1:

**Figure supplement 1.** Contractile properties of the soleus.

Introduction of Errγ into $Mtn^{-/-}$ caused a partial reversal of MHC profile of $Mtn^{-/-}$ towards the WT condition in all muscles examined (*Figure 2C* and *Figure 2—figure supplement 1A and C*). This conversion was only detected within the MHCII subtypes but did not extend to normalization of the proportion of MHCI fibres; in the soleus of $Mtn^{-/-}/Err\gamma^{Tg/+}$, the proportion of MHCIIB fibres was lower than that in $Mtn^{-/-}$ while that of MHCIIA fibres was higher. Nevertheless, both $Mtn^{-/-}$ and $Mtn^{-/-}/Err\gamma^{Tg/+}$ display a lower proportion of MHCI fibres in the soleus muscle than WT (*Figure 2C*).

Next, we examined the mechanism underpinning fibre enlargement. We found that the levels of phosphorylated Akt (an inducer of anabolism) were higher in the muscle of $Mtn^{-/-}$ and $Mtn^{-/-}/Err\gamma^{Tg/+}$ compared to WT (*Figure 2D*). A similar relationship was discovered for its downstream target 4EBP1 (*Figure 2D*). Akt not only promotes protein synthesis but also suppresses catabolism partly by phosphorylating and thereby inactivating FoxO3. We found that deletion of Mtn resulted in an increased ratio of the inactive:active (phosphorylated:non-phosphorylated) form of FoxO3. However, in muscles of $Mtn^{-/-}/Err\gamma^{Tg/+}$ mice the levels of inactive FoxO3 were lower than in that of the $Mtn^{-/-}$ (*Figure 2D*).

## Oxidative fibre profiling and vascular organisation

In all muscles examined, the intensity of the SDH staining (measure of oxidative activity) of fibres was lower in muscle from $Mtn^{-/-}$ compared to WT (*Figure 3A* and *Figure 3—figure supplement 1A–B*). However, upon over-expression of Errγ, the intensity of SDH staining in fibres of $Mtn^{-/-}$ muscle was restored to that of WT. Indeed, also the number of SDH positive fibres was higher than that seen in even the WT muscles albeit not significantly so (*Figure 3A* and *Figure 3—figure supplement 1C*). Introduction of Errγ into $Mtn^{-/-}$ also caused normalization of the number PAS positive fibres (*Figure 3A* and *Figure 3—figure supplement 1D*). The capillary to fibre ratio (C:F); was lowest in the muscles of $Mtn^{-/-}$ mice and highest in those of the $Mtn^{-/-}Err\gamma^{Tg/+}$ mice (*Figure 3B*).

## Metabonomics

The muscle metabolite profile was characterized by [1]H NMR spectroscopy. To identify any metabolic variation driven by the genotypic differences, principal components analysis (PCA) was applied to these profiles. A clear clustering was observed in the scores plot comparing all three genotypic groups demonstrating that they had distinctive metabolite profiles (*Figure 3C*). Comparing the metabolic signature of the $Mtn^{-/-}$ muscle to the $Mtn^{-/-}/Err\gamma^{Tg/+}$ showed clear differences between the two groups (*Figure 3C*) characterised by significantly greater levels of muscle lactate in $Mtn^{-/-}$ muscle compared to that of the $Mtn^{-/-}/Err\gamma^{Tg/+}$ consistent with a greater glycolytic phenotype. Furthermore the levels of creatine/phosphocreatine were also more pronounced in the muscle from $Mtn^{-/-}$ compared to $Mtn^{-/-}/Err\gamma^{Tg/+}$. Errγ modification led to higher taurine and anserine content in the muscle of these animals.

Therefore, histochemical and NMR muscle profiles of the three genotypic groups provide further evidence that Errγ modification of $Mtn^{-/-}$ results in a remodeling of phenotype to a state that differentiates it not only from $Mtn^{-/-}$ but also WT.

## Metabolic gene profile

Key molecular and cellular features that would explain the metabolic profile of $Mtn^{-/-}/Err\gamma^{Tg/+}$ muscle were defined. In the first instance, we examined key regulators of energy metabolism. We found that Errγ over-expression induced changes in levels of two key transcriptional regulators of metabolism; *Perm1* and *Pgc1a* in $Mtn^{-/-}$ muscle (*Figure 4A*).

Next, we examined the expression of key regulators of glucose and fatty acid oxidation (*Glut1, Glut4, Pdk4* and *Had, Lpl* and *Cycs* respectively). We found that *Glut4* and *Pdk4* were lower in $Mtn^{-/-}/Err\gamma^{Tg/+}$ compared to $Mtn^{-/-}$. Moreover, *Had* and *Lpl,* was higher in $Mtn^{-/-}/Err\gamma^{Tg/+}$ than in

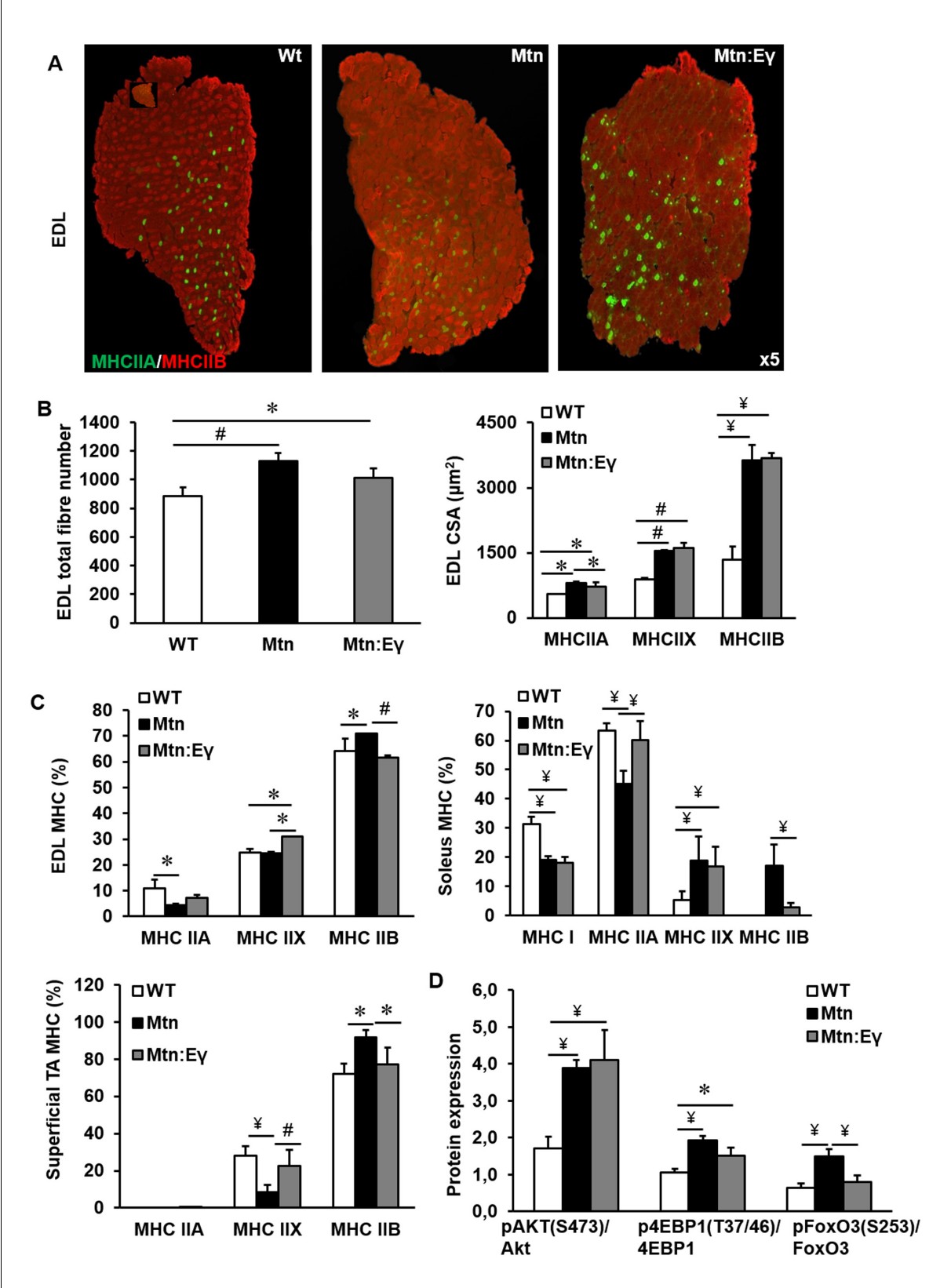

**Figure 2.** Musclespecific expression of ERRγ maintain the hyperplasia in the myostatin null background and normalizes myosin type II phenotype. (A) Representative immunohistochemical images for MHC IIA and IIB staining in the EDL muscle. (B) EDL total fibre number and myofibre cross sectional

*Figure 2 continued on next page*

*Figure 2 continued*

area (CSA, μm$^2$). (**C**) EDL, soleus and superficial TA muscle fibre type composition (**D**) Protein expression of key regulators that control anabolism (pAKT, p4EBP1) and catabolism (pFoxO3) in the gastrocnemius muscle. N = 5 male twelve-week old mice per group; One-way ANOVA followed by Bonferroni's multiple comparison tests, *$p < 0.05$, #$p < 0.01$, ¥$p < 0.001$.

The following figure supplement is available for figure 2:

**Figure supplement 1.** Reprogramming of the soleus myostatin null muscle by ERRγ.

$Mtn^{-/-}$. Of particular note was the finding that the expression of markers of fatty acid metabolism, *Had* and *Lpl*, were not only higher in $Mtn^{-/-}/Err\gamma^{Tg/+}$ than in $Mtn^{-/-}$ but also than in the WT condition (*Figure 4A*).

Oxidative metabolism relies on oxygen that can be stored in muscle by myoglobin. Secondly, oxidative metabolism generates destructive radicals which can be broken down by enzymes including catalase. We found that expression of *myoglobin*, which facilitates the diffusion of oxygen, and *catalase*, an anti-oxidant enzyme, were higher in the muscle of $Mtn^{-/-}/Err\gamma^{Tg/+}$ than in WT and $Mtn^{-/-}$ mice (*Figure 4A*).

Then we investigated genes that control oxidative energetics and examined the expression of molecules controlling fat metabolism (fatty acid transport and uptake molecules: *Cd36*, *Slc25a20*, *Fatp1*, *Fabp3* and regulators of fatty acid oxidation: *Acadl*, *Acadm*). We found that all six genes were expressed to a higher degree in $Mtn^{-/-}/Err\gamma^{Tg/+}$ than in $Mtn^{-/-}$ and WT mice (*Figure 4A*).

We established if the differences in oxidative metabolism between $Mtn^{-/-}$ and $Mtn^{-/-}Err\gamma^{Tg/+}$ in muscle were mirrored by factors related to the microvascular supply to the muscle. We found that the expression of endothelial mitogenic factors (*Vegfa165*, *Vegf189* and *Ffg1*) was lower in the muscles of $Mtn^{-/-}$ than WT mice, but similar in those of $Mtn^{-/-}Err\gamma^{Tg/+}$ and WT mice (*Figure 4B*).

Therefore, the musclespecific expression of Errγ in $Mtn^{-/-}$ mice not only normalizes its metabolic molecular profile but also results in a better microvascular supply of the muscle.

## Ultra-structure

The ultra-structure of muscle in the three cohorts were examined. Using transmission electron microscopy, we found a number of abnormalities in the structure of muscle from $Mtn^{-/-}$ mice heterogeneously sized sarcomeres, misaligned and disrupted Z-Lines, large inter-sarcomeric spaces and altered mitochondrial distribution and size (*Figure 5A*). In contrast, the muscle from $Mtn^{-/-}/Err\gamma^{Tg/+}$ largely lacked these abnormalities (*Figure 5A*). We found that the density of mitochondria in both sub-membrane and intrafusal locations was decreased significantly following the deletion of Mtn. However, the expression of Errγ significantly increased the mitochondrial density at both locations compared to $Mtn^{-/-}$ and at the major site, the sub-membrane region, increased it even compared to WT. Mitochondrial hypertrophy has been postulated to compensate for decreased mitochondrial number or function. Hypertrophy is thought to either protect against apoptosis or for functional mitochondria to fuse with aberrant ones resulting in the maintenance of cell function (*Frank et al., 2001*; *Ono et al., 2001*). Mitochondrial hypertrophy was evident in both compartments in muscle from $Mtn^{-/-}$ (*Figure 5B–E*) and was normalized by Errγ in the sub-membrane region (*Figure 5D*).

These results show that the deletion of Mtn leads to numerous ultra-structural abnormalities. Over-expression of Errγ in the $Mtn^{-/-}$ prevents almost all the ultra-structural abnormalities.

## Myonuclear organization and satellite cell

We next examined the features of individual muscle fibres to determine the effect of Errγ in $Mtn^{-/-}$ mice. We found, that deletion of Mtn resulted in fewer satellite cells compared to WT and that the number of satellite cells was even lower in the muscles of the $Mtn^{-/-}/Err\gamma^{Tg/+}$ mice (*Figure 6A,C and D*). Next, we determined proliferation and differentiation characteristics of satellite cells in the three cohorts. We found that following 48 hr of culture, the number of progeny had increased in all the genotypes but the proportional relationship found in uncultured fibres persisted (*Figure 6E–F*). During the 48 hr period of culture, satellite cells not only divide but also form clusters (*Figure 6G–H*). We found that the number of clusters were similar in fibres from WT and $Mtn^{-/-}$ (*Figure 6G*), but

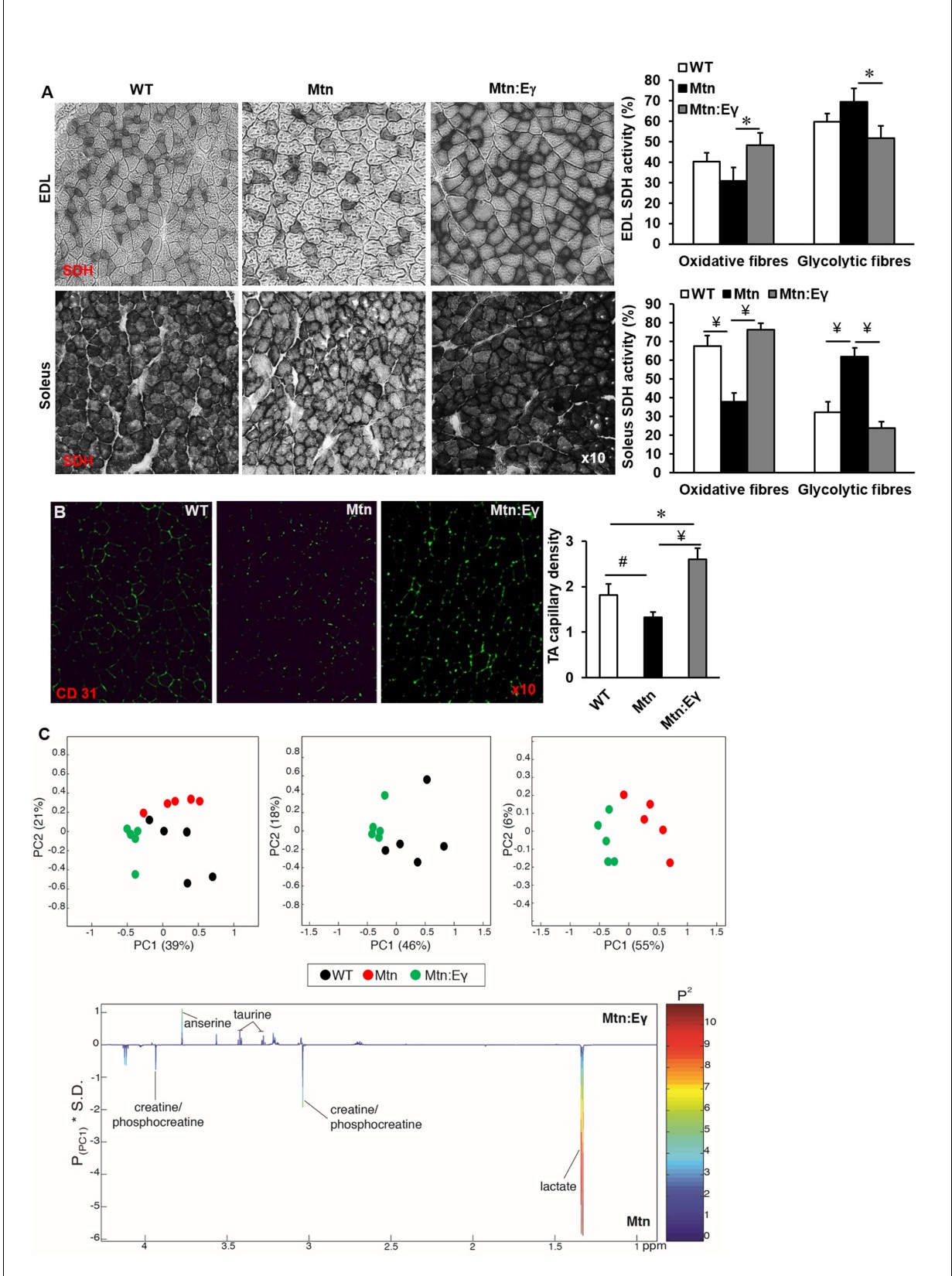

**Figure 3.** Musclespecific expression of ERRγ normalizes the metabolic and capillary profile of myostatin null mice. (**A**) SDH staining and quantification of EDL and soleus muscles of Wt, *Mtn* and *Mtn:Eγ* mice. N = 5 male twelve-week old mice per group; One-way ANOVA followed by Bonferroni's

*Figure 3 continued on next page*

*Figure 3 continued*

multiple comparison tests, *p<0.05, #p<0.01, ¥p<0.001. (**B**) Muscle capillary density as determined by CD31 staining. (**C**) Pair-wise comparisons of the metabolic profiles obtained from the gastrocnemius muscle from WT, *Mtn* and *Mtn:Eγ* mice. Principal components analysis (PCA) scores plots comparing WT, *Mtn* and *Mtn:Eγ*; WT and *Mtn:Eγ*; as well as *Mtn* and *Mtn:Eγ*); (% variance in the parenthesis). Colour loadings plots shown for PC1 of the model comparing *Mtn* and *Mtn:Eγ*. Product of PC loadings with standard deviation of the entire data set coloured by the square of the PC loading.

The following figure supplement is available for figure 3:

**Figure supplement 1.** Reprogramming of the tibialis anterior muscle of myostatin null mice by ERRγ.

there were fewer clusters in the $Mtn^{-/-}/Err\gamma^{Tg/+}$-derived cultures. The number of cells per cluster was highest in WT and lowest in the $Mtn^{-/-}$ with that of the $Mtn^{-/-}/Err\gamma^{Tg/+}$ in between the two (*Figure 6H*). Finally, we found deletion of Mtn and the introduction of Errγ did not impact on the process of differentiation (*Figure 6I*).

Myonuclear number and organization were then determined. First, there were significantly more myonuclei in the fibres of $Mtn^{-/-}/Err\gamma^{Tg/+}$ compared to WT (*Figure 6A–B*). Secondly, we examined the distribution of myonuclei within a fibre. This is thought to be a regulated process since myonuclei position is important to minimize issues related to macromolecule movement in larger cells. Therefore the degree of regulation is inversely proportional to random positioning of the nuclei (*Bruusgaard et al., 2003*). In order to quantify this, we calculated the distance to the nearest

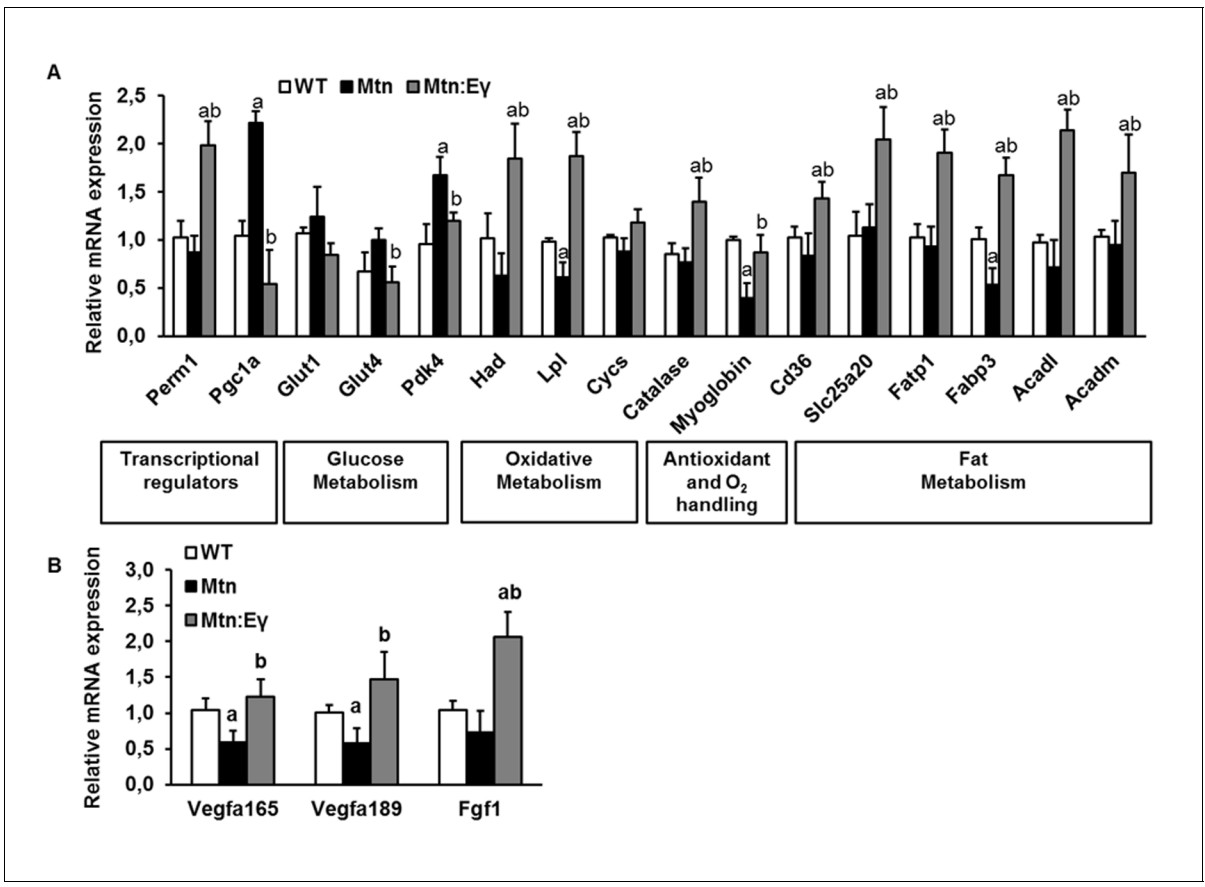

**Figure 4.** Molecular reprogramming of myostatin null muscle by ERRγ and its ability to promote capillary formation by the expression of angiogenic factors. (**A**) Gene expression levels of transcriptional regulators, glucose metabolism regulators, oxidative metabolism genes, antioxidant and oxygen handling genes and fat metabolism genes. (**B**) Angiogenic gene expression. 'a' denotes changed significantly from WT and 'b' denotes changes significantly from *Mtn*. N = 5 male twelve-week old mice per group; One-way ANOVA followed by Bonferroni's multiple comparison tests, p<0.05.

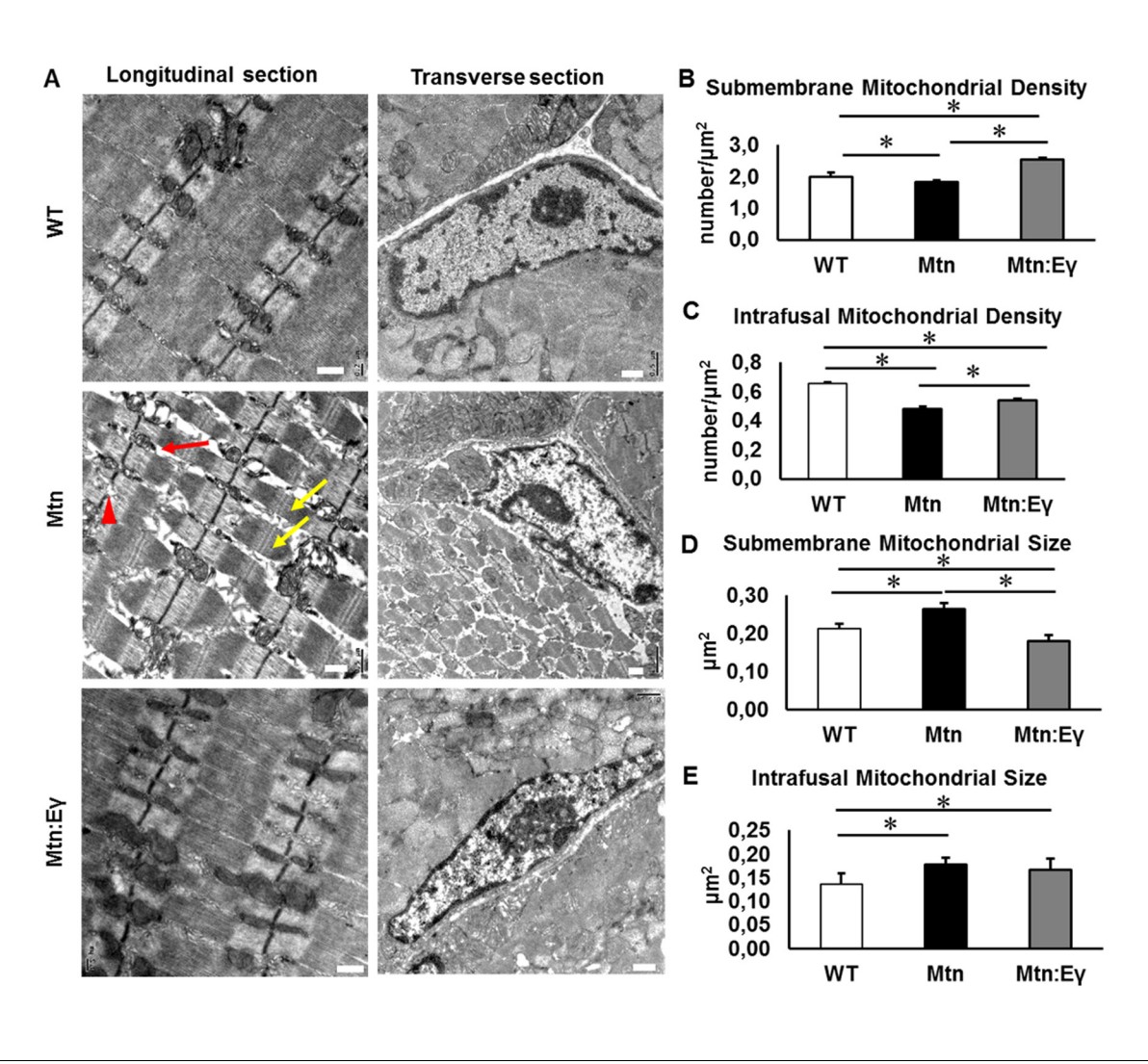

**Figure 5.** Musclespecific expression of ERRγ normalizes ultra-structural abnormalities myostatin null mice. (A) Transmission electron microscopy images in longitudinal and transverse sections of WT, *Mtn* and *Mtn:Eγ* muscle, scale 0.5 μm. Note the large spaces (red arrow) disrupted Z-lines (red arrowhead) and non-uniform sarcomere width (yellow arrows). (B) Quantification of submembrane mitochondrial density. (C) Quantification of Intrafusal mitochondrial density. (D) Quantification of submembrane mitochondrial size. (E) Quantification of intrafusal mitochondrial size. N = 3 male twelve-week old mice per group; One-way ANOVA followed by Bonferroni's multiple comparison tests, *p<0.05.

neighbour for the nuclei located at the periphery of single fibres from WT, $Mtn^{-/-}$ and $Mtn^{-/-}/Err\gamma^{Tg/+}$ mice. Confocal stacks of single fibres labelled with DAPI (*Figure 6J*) were used to generate the 3D coordinates of each nucleus in a fibre (*Figure 6K*) using Imaris software. Using custom made software, a simulation of randomly and optimally distributed nuclei was compared to the actual distribution (see Materials and methods). The WT fibres displayed an improvement from a random distribution of 20%. However $Mtn^{-/-}$ and $Mtn^{-/-}/Err\gamma^{Tg/+}$ fibres had distributions that were more random, with significantly lower improvements of 10% and 4%, respectively (*Figure 6L*). These results show that the expression of Errγ in the $Mtn^{-/-}$ does not normalize key features related to either the satellite cells, myonuclei number of their positioning.

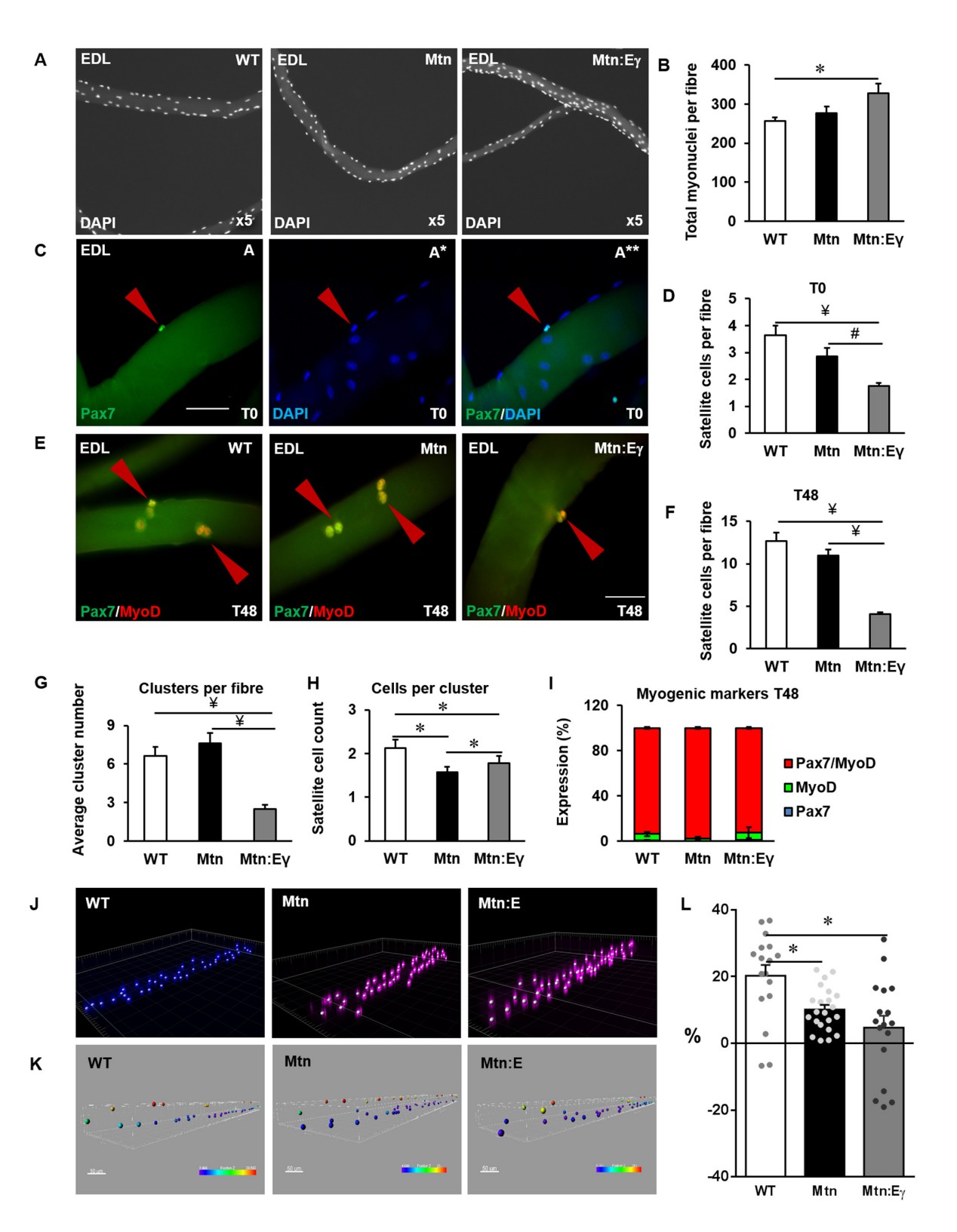

**Figure 6.** Oxidative muscle developed through ERRγ in the muscle of myostatin null mice shows depletion of satellite cells and increased myonucle content. (A) Single EDL muscle fibres stained with DAPI to visualize myonuclei. (B) Quantification of myonuclear number in EDL fibres. (C) Quiescent

*Figure 6 continued on next page*

*Figure 6 continued*

satellite cells stained for Pax7 on freshly isolated (T = 0 hr) muscle fibres from the EDL (arrowhead). (D) Quantification of satellite cell number on freshly isolated EDL fibres. (E) Single muscle fibres after 48 hr in cell culture stained for DAPI, Pax7 and MyoD (arrowhead). (F) Quantification of total number of cells on muscle fibre at 48 hr. (G) Quantification of satellite cell clusters at 48 hr. (H) Cluster size at 48 hr on muscle fibres. (I) Profiling of differentiation at 48 hr. (J) Confocal stacks of single fibres labelled with DAPI to study myonuclear organization. (K) Virtual reconstruction of single muscle fibres, colour encodes distance in the z-plane. (L) Improvement in myonuclear organization, where 0% denote a random distribution. Fibres were from 4 male twelve-week old mice per group; One-way ANOVA followed by Bonferroni's multiple comparison tests, *p<0.05, #p<0.01, ¥p<0.001.

## Skeletal muscle regeneration

Thus, far all the changes in muscle resulting from the over–expression of Errγin the$Mtn^{-/-}$ were beneficial except for a lower number of satellite cells. In this section we determined the consequence of this deficit on the ability of skeletal muscle to regenerate, a process reliant on satellite cells. To that end, we induced injury of the TA using cardiotoxin and the progression of regeneration assessed at three crucial time points; day three (D3) as the process of debris clearance is ongoing and regeneration of fibres begins, day six (D6) when robust fibre regeneration can be quantified and day fourteen (D14) when debris clearance has been completed.

At D3 the muscle clearance of dying fibres was slowest in $Mtn^{-/-}$ compared to the other two genotypes (*Figure 7A–B*). Clearance is mediated in part by macrophages and we found that the density of macrophages was highest in the muscle of $Mtn^{-/-}/Err\gamma^{Tg/+}$ compared to either $Mtn^{-/-}$ or WT (*Figure 7C–D*). Furthermore, we found that the TA from $Mtn^{-/-}/Err\gamma^{Tg/+}$ at the early stages of generation contained the highest number of committed muscle cells (*Figure 7E–F*).

By D6, there was a greater degree of regeneration (size of newly formed eMHC⁺ fibres) in $Mtn^{-/-}/Err\gamma^{Tg/+}$ compared to either $Mtn^{-/-}$ or WT (*Figure 7G–H*) and a more advanced removal of dying fibres in $Mtn^{-/-}/Err\gamma^{Tg/+}$ than in $Mtn^{-/-}$ (*Figure 7I–J*). We also found evidence for a lower amount of cell death in the regenerating areas of $Mtn^{-/-}/Err\gamma^{Tg/+}$ than $Mtn^{-/-}$ or WT mice (*Figure 7K–L*). At D6 macrophage activity was still high in the muscle of $Mtn^{-/-}/Err\gamma^{Tg/+}$ compared to either $Mtn^{-/-}$ or wild type (*Figure 7M–N*) as were the number of committed (Myo⁺/Pax7⁻) muscle progenitor cells (*Figure 7O–P*). Precocious differentiation could lead to an exhaustion of cells which would ultimately attenuate fibre size. To examine this line of thought we examined damaged muscles at an advanced stage of regeneration (D14). We found further evidence for accelerated regeneration in the $Mtn^{-/-}/Err\gamma^{Tg/+}$ compared to either $Mtn^{-/-}$ or WT gauged by a decrease in the density of fibres still expressing eMHC (*Figure 7—figure supplement 1A*). Importantly, there was no deficit in the size of newly regenerated fibre in the muscle of $Mtn^{-/-}/Err\gamma^{Tg/+}$ mice (*Figure 7—figure supplement 1B*). These results show that even though the muscles of $Mtn^{-/-}/Err\gamma^{Tg/+}$ have fewer satellite cells than the muscles of the WT and $Mtn^{-/-}$ mice, their muscle regenerating capacity exceeds that of both $Mtn^{-/-}$ and WT mice.

## Non-genetic post-natal induction of oxidative skeletal muscle growth

Our newly generated hypermuscular, hyper-oxidative mouse line ($Mtn^{-/-}/Err\gamma^{Tg/+}$) displays a number of characteristics that make them attractive both in terms of physiology and regeneration. However, the muscle phenotype in these models is largely established during embryonic and post-natal development. Therefore, we next established if similar phenotypes could be obtained via non-genetic modifications. To do so, we inhibited Mtn at post-natal stages in $Err\gamma^{Tg/+}$ mice (which displays an increased oxidative profile) by weekly injections of soluble activin receptor IIB protein (sActRIIB), which has been shown to antagonize signalling mediated by myostatin and related-proteins.

Following 8 weeks of weekly injections we found that sActRIIB caused an increase in the body mass of both WT and $Err\gamma^{Tg/+}$ mice (*Figure 8A*). Examination of isolated muscles showed an increase in muscle mass of approximately 70% in the EDL of WT and 44% in $Err\gamma^{Tg/+}$ above age-matched control animals (*Figure 8B*). Other muscles examined showed a similar increase in muscle mass (*Figure 8—figure supplement 1A*). The increase in muscle mass was not due to an increase in fibre number (data not shown) but due to hypertrophy of all MHC fibre types (*Figure 8—figure supplement 1B*). There was no change in the MHC fibre type composition following the injection of sActRIIB in either genotype (*Figure 8C and F*). However, we found that injection of sActRIIB induced a decrease in the oxidative capacity of the muscle in WT mice as indicated by a decreased proportion

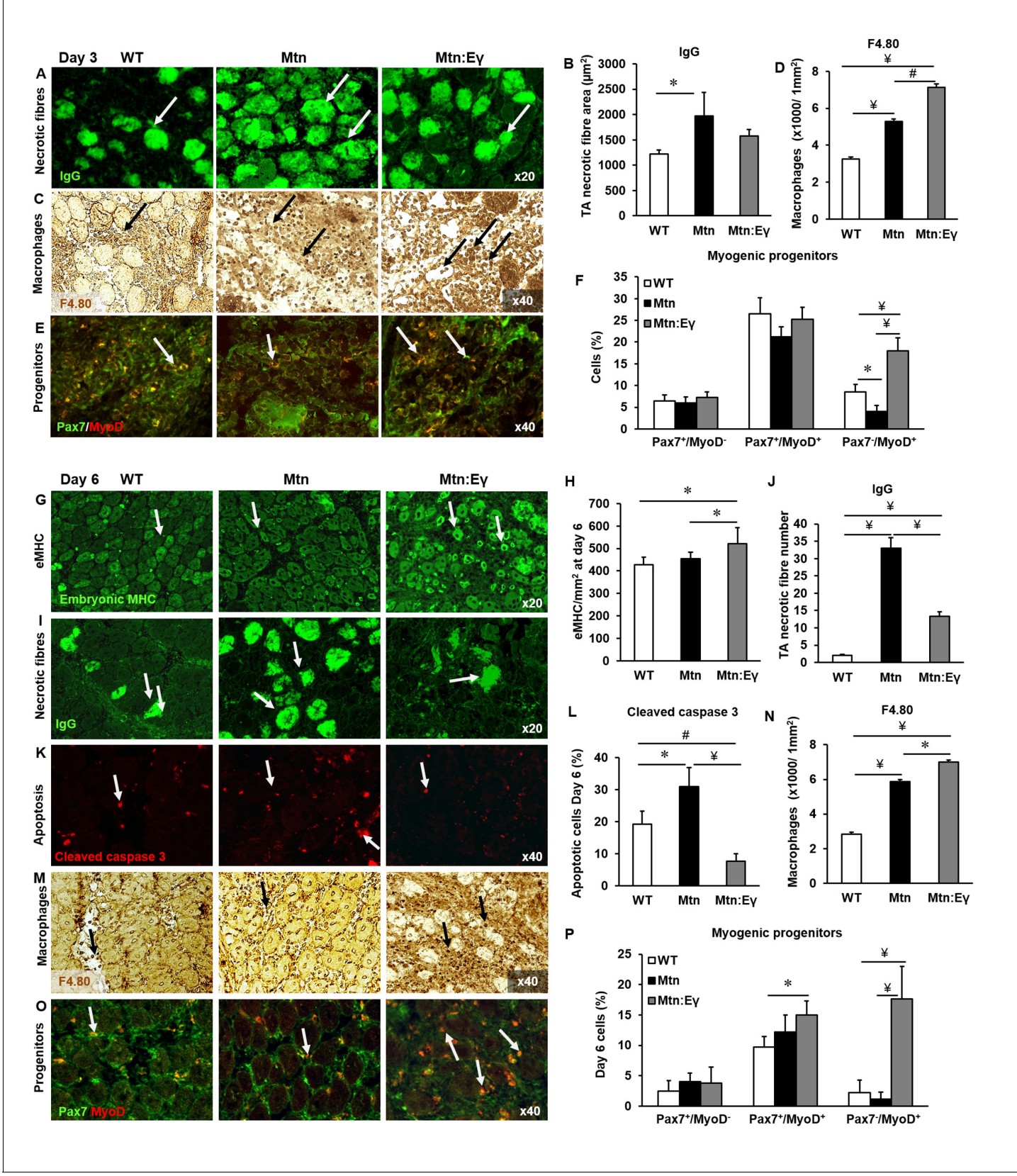

**Figure 7.** Skeletal muscle regeneration is accelerated by the expression of Errγ in myostatin null mice through enhanced macrophage and satellite cell activity. Skeletal muscle regeneration in response to cardiotoxin injury. (**A**) Muscle necrotic fibres visualized by IgG staining at Day 3 (arrows). (**B**)
*Figure 7 continued on next page*

*Figure 7 continued*

Quantification of dying fibre size at Day 3. (**C**) Macrophage infiltration in the TA muscle using an F4.80 antibody at Day 3 (arrows). (**D**) Quantification of macrophage density in damaged muscle. (**E**) Myogenic progenitors at Day 3. Pax-7 detection in green, MyoD expressing cells in red (arrows). (**F**) Quantification of uncommitted muscle cells (Pax-7$^+$/MyoD$^-$), precursor (Pax-7$^+$/MyoD$^+$) and committed (Pax-7$^-$/MyoD$^+$) muscle cells at Day 3. (**G**) Expression of embryonic myosin heavy chain on Day 6 (arrows). (**H**) Quantification of regenerating muscle fibres at Day 6. (**I**) Necrotic fibres at Day 6 detected via infiltrated fibre IgG profiling (arrows). (**J**) Quantification of dying muscle fibres at Day 6. (**K**) Cleaved caspase 3 staining at Day 6 as a marker of apoptosis (arrows). (**L**) Quantification of apoptotic density at Day 6. (**M**) Macrophage infiltration in the TA on Day 6 (arrows). (**N**) Quantification of macrophage infiltration at Day 6. (**O**) Myogenic progenitors on Day 6. Pax-7 detection in green, MyoD expressing cells in red (arrows). (**P**) Quantification of uncommitted muscle cells (Pax-7$^+$/MyoD$^-$), precursor (Pax-7$^+$/MyoD$^+$) and committed (Pax-7$^-$/MyoD$^+$) muscle cells at Day 6. N = 4/5 male twelve-week old mice per group; One-way ANOVA followed by Bonferroni's multiple comparison tests, $*p<0.05$, $\#p<0.01$, ¥$p<0.001$.

The following figure supplement is available for figure 7:

**Figure supplement 1.** Characterisation of regenerating tibialis anterior muscle at day 14.

---

of SDH$^+$ fibres (*Figure 8D and G*). Strikingly, sActRIIB did not cause a reduction in the proportion of SDH$^+$ fibres in *Errγ$^{Tg/+}$* (*Figure 8D and G*). The increased oxidative capacity of the muscle was accompanied with a rise in the number of capillaries serving each fibre in the muscle from *Errγ$^{Tg/+}$* but not WT mice (*Figure 8E and H* and *Figure 8—figure supplement 1D*). These results show that it is possible to induce substantial muscle enlargement while maintaining oxidative capacity, challenging the generally accepted dogma that the size and oxidative capacity of a fibre are, because of diffusion constraints, inversely related.

## Discussion

The main observations of this study are firstly that substantial hypertrophy can occur without a concomitant reduction in fibre oxidative capacity. This observation challenges the dogma that there is a trade-off between muscle fibre size and oxidative capacity. Secondly, our results challenge the notion that slow oxidative muscle has a higher number of satellite cells than those that are fast glycolytic.

A number of studies have shown that deletion of myostatin leads to the development of hypertrophic muscle. Although such enlarged muscles appear essentially normal at the histological level, their ability to generate tension is impaired, particularly during prolonged periods of work (*Amthor et al., 2007*; *Mendias et al., 2006*; *Relizani et al., 2014*). The higher than normal fatigability of the muscle could be attributable to the lower number of mitochondria consequent to deletion of myostatin in the germline (*Amthor et al., 2007*).

To alleviate this mitochondrial deficit in *Mtn$^{-/-}$* mice, we introduced the expression of Errγ into skeletal muscle. This gene is highly expressed in tissues with a high oxidative capacity, such as the heart, kidneys, brain and slow oxidative skeletal muscle where it has been demonstrated to trigger mitochondrial biogenesis (*Hong et al., 1996*; *Heard et al., 2000*; *Giguère, 2008*; *Narkar et al., 2011*). Introduction of Errγ overexpression that would increase oxidative capacity on a *Mtn$^{-/-}$* background that is associated with hypertrophy would challenge the trade-off that is thought to exist between oxidative capacity and fibre size (*Van Der Laarse et al., 1997*; *Degens, 2012*).

One of the key features of *Mtn$^{-/-}$* muscle is the lower SDH activity, indicative of a low oxidative status. This combination of a low oxidative capacity and a large fibre size fits nicely with the concept of the trade-off between fibre size and oxidative capacity. It also is associated with a larger proportion of type IIB fibres than seen in muscles from WT mice. Here we show that even though the muscle mass and fibre sizes did not differ between *Mtn$^{-/-}$* and *Mtn$^{-/-}$/Errγ$^{Tg/+}$* mice, the latter had a higher SDH activity.

The higher SDH activity in *Mtn$^{-/-}$/Errγ$^{Tg/+}$* than *Mtn$^{-/-}$* mice was associated with a partial normalisation of the MHC fibre profile; a decrease in the proportion of IIB fibres in all muscles examined. What was conspicuous, however, was the absence of normalization of the proportion of MHC I fibres. We believe that this is significant and reveals a key feature of the influence of a metabolic programme on muscle physiology. We suggest that the oxidative programme, here driven by Errγ, readily converts IIB to IIA fibres but is that it is unable to induce the transition to type I MHC

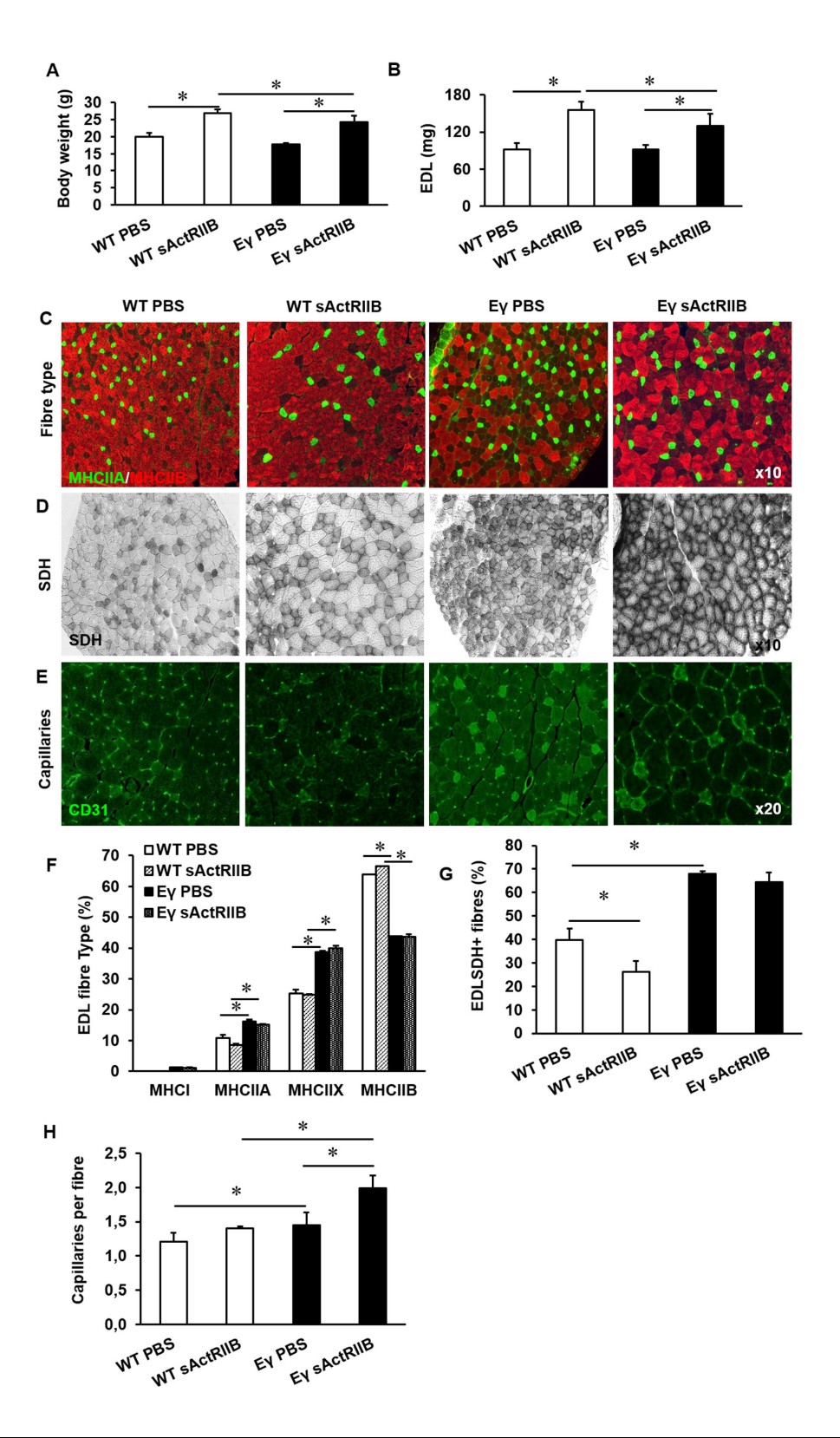

**Figure 8.** Post-natal inhibition of myostatin in the muscle-specific *ERRγ* mice leads to hypertrophic muscle with enhanced oxidative and vascular features. (A) Body mass in 12-week-old mice after an 8 week treatment regime. (B) EDL muscle mass after sACtRIIB treatment. (C) Muscle fibre type

*Figure 8 continued on next page*

*Figure 8 continued*

profiling with MHCIIA (green) and MHCIIB (red) antibodies. (D) Oxidative enzyme profiling using SDH histochemistry. (E) Muscle capillary density profiling with CD31 antibody. Quantification of (F) MHC EDL fibre type, (G) SDH positive fibres, (H) capillary density. Intrafibre staining in the *Errγ* muscle in (D) is artefact and was ignored in all quantification procedures; One-way ANOVA followed by Bonferroni's multiple comparison tests, *$p < 0.05$.

The following figure supplement is available for figure 8:

**Figure supplement 1.** Muscle characterisation after post-natal inhibition of myostatin in the muscle specific *ERRγ* mice.

isoforms. Energy status (ATP/ADP or phosphocreatine) has been implicated as a determinant of the MHC fibre type with high levels inducing ever more fast forms in keeping with their myofibrillar ATPase activity (*Conjard et al., 1998*; *Bottinelli et al., 1994*). We show here from our NMR profiling that indeed the muscle of $Mtn^{-/-}$ has high levels of phosphocreatine, which would be in keeping with the high ATPase activity of Type IIB fibres found in its muscle. Furthermore, we show Errγ over-expression in the muscle of $Mtn^{-/-}$ normalizes this feature yet does not lead to the formation of I fibres. This observation adds to a growing body of evidence that the type II programme is plastic and adaptable whereas the Type I fibres are more resistant to change (*Sutherland et al., 1998*) and may not be part of the IIB⟷IIX⟷IIA continuum. Indeed a number of studies have questioned whether the 'final step' (conversion of Type IIA to I) is even possible. Development of type I fibres has been described in a number of conditions, for example following Chronic low-frequency stimulation (CLFS) (*Peuker et al., 1999*; *Kwong and Vrbová, 1981*). However, these studies never examined whether Type I were formed as a consequence of the remodeling of Type II fibres or through the formation of new fibres, a process that would require satellite cells. Indeed the development of Type I fibres following extended CLFS can only be induced to significant levels when accompanied by robust myofibre regeneration (*Pette et al., 2002*; *Maier et al., 1988*). Taken together, these studies imply that myostatin signalling acts at an embryonic/foetal stage of muscle development to pattern a subpopulation of satellite cells/muscle precursors in a muscle specific manner to form Type I fibres. The protocol of over-expressing Errγ used in this study is unable to influence this process.

One of the intriguing aspects of the $Mtn^{-/-}$ phenotype is the concurrence of a larger muscle mass and a low oxidative capacity, as also reflected by a low mitochondrial content (*Amthor et al., 2007*). As mentioned above, this association corresponds with the prediction of the concept of a trade-off between muscle fibre size and oxidative capacity. There could, however, also be another function for the high glycolytic capacity. For instance, the Warburg Effect is the observation that most cancer cells rely on glycolysis even in the presence of oxygen (*Warburg et al., 1927*) for the production of a intermediates essential for the building blocks of any cell including nucleic acids, lipids and proteins (*Deberardinis et al., 2008*). In a similar way, glycolysis in the muscles of $Mtn^{-/-}$ mice may support the high levels of protein synthesis required for the initial muscle hypertrophy and maintenance of the large muscle mass. An interesting point is that such cells are not only dependent on glycolysis but also often have decreased oxidative phosphorylation capacity (*Petros et al., 2005*). Where the similarities between the Warburg Effect in cancer cells and findings from this study differ is the outcome following an intervention that promotes oxidative metabolism. In cancer cells such an intervention reduces cell growth (*Wang and Moraes, 2011*) while we have shown with Errγ overexpression on a $Mtn^{-/-}$ background not only re-establishes the oxidative capacity but also maintains the hypertrophic state. Consistent with the oxidative metabolic phenotype of the $Mtn^{-/-}/Errγ^{Tg/+}$ mice are the higher levels of taurine and anserine observed in the NMR metabonomic analysis, since taurine is positively correlated with the oxidative capacity of muscle tissues (*Dunnett et al., 1997*). Anserine is β-alanine and histidine related dipeptide with antioxidant properties commonly found in skeletal muscle of many animals (*Kohen et al., 1988*). Thus, it may act as a scavenging agent of the byproducts arising from elevated oxidative activity in the muscle of $Mtn^{-/-}/Errγ^{Tg/+}$ mice.

A number of studies have suggested that fibres that rely on oxidative phosphorylation limit their size in order that oxygen from the capillaries diffuses efficiently into the cells and to the mitochondria for ATP production (*Kinsey et al., 2007*; *Van Der Laarse et al., 1997*; *van Wessel et al., 2010*). The large fibres with a low oxidative capacity in $Mtn^{-/-}$ mice conform to this concept and have a low capillary supply per fibre. During compensatory hypertrophy the time course of angiogenesis

and fibre hypertrophy are similar (*Egginton et al., 1998*; *Plyley et al., 1998*) and the capillary supply to a fibre is related to the size of the fibre (*Ahmed et al., 1997*; *Degens et al., 1994*). Such a coupling between the fibre size and capillary supply seems to be altered in the $Mtn^{-/-}$ mice in such a way that they have fewer capillaries than expected for the size of the fibre. However, over-expression of Errγ in either WT or $Mtn^{-/-}$ drives a robust angiogenic gene programme, increases the number of capillaries per fibre and ultimately muscle blood flow as shown previously (See *Figure 3B* and (*Narkar et al., 2011*; *Matsakas et al., 2012b*). An important finding here is that the angiogenesis programme promoted by muscle expression of Errγ is responsive to change in fibre size so that when a fibre grows, it stimulates the formation of blood vessels presumably to ensure optimal perfusion (*Figure 8H*). Two additional modifications take place, an increase in myoglobin transcription and increasing the density of mitochondria at the sarcolemma that would sustain large oxidative fibres developed as a consequence of Errγ in the $Mtn^{-/-}$ background. These outcomes have been postulated to prevent a decline in maximum steady state power as an oxidative fibre increases size (*Hickson, 1980*; *Heard et al., 2000*).

In this study, we show that the muscle hypertrophy that develops following germline deletion of Mtn has many ultrastructural abnormalities including splitting of sarcomeres, misaligned Z-lines and alteration in mitochondrial distribution and morphology. The maintenance of muscle structure is largely mediated by mechanisms that remove unwanted proteins and organelles through either the proteasome or autophagic pathways (*Sandri, 2013*; *Bonaldo and Sandri, 2013*). Furthermore, deregulated proteasome activity or autophagy leads to muscle wasting in a number of diseased conditions (*Sandri et al., 2004*; *Carmignac et al., 2011*). As these pathways are involved in anabolic processes, it seems intuitive that they should be tuned down in order to support muscle growth. Indeed, we show that the activity of a key regulator of these processes, FoxO3 is suppressed in the absence of Mtn. However, we show that Errγ expression in muscle leads to a substantial normalization of the ultrastructure $Mtn^{-/-}$ skeletal muscle as well an improvement in the specific force. Most importantly, we show that a more physiological measure of muscle function- fatigability, is not only normalized but exceeds the value of WT mice. Our data demonstrate that the suppression of FoxO3 activity is alleviated by Errγ. We suggest that the molecular and organelle clearance programmes being mediated by FoxO3 are generally not anabolic but are rather there to maintain cellular homeostasis. However, when its activity is attenuated, it leads to an accumulation of structural abnormalities that compromises muscle function. Nevertheless, not all features of the $Mtn^{-/-}$ muscle were normalised by Errγ expression; Myonuclei in $Mtn^{-/-}$ and $Mtn^{-/-}/Errγ^{Tg/+}$ were more disorganized than those in WT fibres. Proper nuclear positioning is probably required for normal muscle function, possibly due to irregular size and spacing of myonuclear domains (*Metzger et al., 2012*) and myonuclear disorganization is observed both in ageing skeletal muscle and in models of muscular dystrophies (*Bruusgaard et al., 2006*; *Meinke et al., 2014*). Additionally, accretion of myonuclei is a prerequisite for maintaining specific force during hypertrophy and mitochondrial protein systems have been suggested to play a role in defining myonuclear domain size in rodents (*Liu et al., 2009*). The increased number of myonuclei and increased synthesis of mitochondria in the $Mtn^{-/-}/Errγ^{Tg/+}$ mice might compensate for the observed disorganized myonuclei, restoring specific force and ultrastructure.

Finally, our study gives a new perspective on the relationship between metabolism, satellite cell numbers and their activity during regeneration. A number of studies have implied that slow muscles contain more satellite cells than fast (*Putman et al., 1999*; *Christov et al., 2007*). In this study, we show that at least in the EDL as the fibres transitioned from Type IIB to Type IIA, the number of associated satellite cells was significantly reduced. One possible explanation for this finding is by taking into account the concomitant increase in the number of nuclei in the myofibre. Here, the relationship is opposite to satellite cell fibre number. We postulate that the absence of myostatin promotes myoblast fusion at the expense of satellite cell. Furthermore, that over-expression of Errγ exacerbates this relationship. Severe depletion of satellite cell numbers has been reported to severely retard the process of muscle regeneration (*Schuster-Gossler et al., 2007*; *Vasyutina et al., 2007*). Here, we show that the depletion of satellite cells to less than 50% of their normal levels does not impact on skeletal muscle regeneration since they have a vast capacity to generate precursors which in most situations are never realized fully (*Collins et al., 2005*). Instead, we suggest that oxidative environment established by Errγ is the key determinant in accelerating regeneration. Our work supports previous work showing that oxidative metabolism supports muscle regeneration (*Lowrie et al.,*

*1982*; *Matsakas et al., 2012b*, *2013*) and are in agreement with a number of studies showing that genetic manipulations leading to a greater oxidative capacity accelerate muscle regeneration (*Li et al., 2007*; *Hussain et al., 2013*). One possible explanation for our results is our finding that Errγ promotes hyper-vascularization. Angiogenesis is a key determinant in the muscle regeneration process. We suggest that the reduction of satellite cell is off-set by the ability to promote vascularization and clearance of the necrotic tissues, allowing the small number of satellite cells to expand greatly to enact rapid repair. This hypothesis is supported by our data investigating both macrophage density and the generation of myoblast in the $Mtn^{-/-}/Err\gamma^{Tg/+}$ mice. Many studies have found that programmes of muscle repair are often at the expense of satellite cells which are not available for future cycles of degeneration/regeneration (*Castets et al., 2011*). We will investigate this avenue of research in the future by conducting a second round tissue damage in the three genetic lines described here. Encouragingly, our data show that although there was an increase in the number of myogenic precursors ($Pax7^+/MyoD^+$) as well as committed cells ($Pax7^-/MyoD^+$) in the $Mtn^{-/-}/Err\gamma^{Tg/+}$ compared to WT at D6, this was not at the expense of cells with satellite cell character ($Pax7^+/MyoD^-$).

In summary, our work challenges the dogma of an inverse relationship between muscle fibre size and oxidative capacity. The deviation from this relationship may be realized by the increased capillarisation and myoglobin content of the muscle and redistribution of mitochondria to a subsarcolemmal location. These adaptations were not associated with the loss of muscle force generating capacity and in fact even resulted in improved exercise capacity. It is likely that the increased microvascular network plays a crucial role in muscle regeneration as the $Mtn^{-/-}/Err\gamma^{Tg/+}$ mice had even lower satellite cell numbers than $Mtn^{-/-}$ mice, yet a regenerative capacity that even exceeded that of WT mice. In future we will determine whether it confers other advantages in particular the ability to confer resistance to obesity and sarcopenia.

## Materials and methods

### Ethical approval

The experiments were performed under a project license from the United Kingdom Home Office in agreement with the Animals (Scientific Procedures) Act 1986. The University of Reading Animal Care and Ethical Review Committee approved all procedures. Animals were humanely sacrificed via Schedule 1 killing between 8:00–13:00.

### Animal maintenance

Healthy C57Bl/6 (WT), $Mtn^{-/-}$, $Mtn^{-/-}/Err\gamma^{Tg/+}$ and $Err\gamma^{Tg/+}$ mice were bred and maintained in accordance to the Animals (Scientific Procedures) Act 1986 (UK) and approved by the University of Reading in the Biological Resource Unit of Reading University. Mice were housed under standard environmental conditions (20–22°C, 12–12 hr light–dark cycle) and provided food and water *ad libitum*. We used male mice that were 4–5 months old at the start of the study. Each experimental group consisted of 3–12 mice. $Mtn^{-/-}$ and $Err\gamma^{Tg/+}$ mice were a gift of Se-Jin Lee (John's Hopkins USA) and Ronald Evans respectively (Salk Institute for Biological Studies, La Jolla, USA). Post-natal muscle growth was induced in one month-old males WT and $Err\gamma^{Tg/+}$ mice that were injected twice weekly intraperitoneally with 10 mg/kg of the soluble activin receptor IIB (sActRIIB-Fc) for a period of two months. Each experimental group consisted of 5–6 mice.

### Exercise fatigue test

Mice were acclimatised to running on a treadmill in three sessions (10 m·min$^{-1}$ for 15 min followed by a 1 m·min$^{-1}$ increase per minute to a maximum of 12 m·min$^{-1}$) (Columbus Instruments Model Exer 3/6 Treadmill, Serial S/N 120416). Exhaustion was determined by exercising the mice at 12 m·min$^{-1}$ for 5 min, followed by 1 m·min$^{-1}$ increases to a maximum of 20 m·min$^{-1}$ until the mouse was unable to run.

### Muscle tension measurements

Dissection of the hind limb was carried out under oxygenated Krebs solution (95% $O_2$ and 5% $CO_2$). Under circulating oxygenated Krebs solution one end of a silk suture was attached to the distal

tendon of the extensor digitorum longus (EDL) and the other to a Grass Telefactor force transducer (FT03). The proximal tendon remained attached to the tibial bone. The leg was pinned to a Sylgard-coated experimental chamber. Two silver electrodes were positioned longitudinally on either side of the EDL. A constant voltage stimulator (S48, Grass Telefactor) was used to directly stimulate the EDL which was stretched to attain the optimal muscle length to produce maximum twitch tension ($P_t$). Tetanic contractions were provoked by stimulus trains of 500 ms duration at, 10, 20, 50, 100 and 200 Hz. The maximum tetanic tension ($P_o$) was determined from the plateau of the frequency-tension curve. Specific force was estimated by normalising tetanic force to EDL muscle mass (g).

## Histological analysis and immunohistochemistry

Following dissection, the muscle was immediately frozen in liquid nitrogen-cooled isopentane and mounted in Tissue Tech freezing medium (Jung) cooled by dry ice/ethanol. Immunohistochemistry was performed on 10 μm cryosections that were dried for 30 min before the application of block wash buffer (PBS with 5% foetal calf serum (v/v), 0.05% Triton X-100). Antibodies were diluted in wash buffer 30 min before use. Details of primary and secondary antibodies are given in *Supplementary file 1*. F4/80 was detected using the Vector Laboratories ImmPRESS Excel Staining Kit. Morphometric analysis of fibre size was performed as previously described (*Matsakas et al., 2012a*).

## Succinate dehydrogenase (SDH) staining

Transverse EDL muscle sections were incubated for 3 min at room temperature in a sodium phosphate buffer containing 75 mM sodium succinate (Sigma), 1.1 mM Nitroblue Tetrazolium (Sigma) and 1.03 mM Phenazine Methosulphate (Sigma). Samples were then fixed in 10% formal-calcium and cleared in xylene prior to mounting with DPX mounting medium (Fisher). Densitometry of the samples was performed on a Zeiss Axioskop2 microscope mounted with an Axiocam HRc camera. Axiovision Rel. 4.8 software was used to capture the images.

## Transmission electron microscopy

To identify the distribution of the mitochondria in the muscle fibres, biceps and extensor carpi radialis muscle were removed cut in pieces of 1 mm$^3$ and immerse fixed in 4% PFA and 2.5% glutaraldehyde in 0.1 M cacodylate buffer pH 7.4 (4°C, 48 hr). Tissue blocks were contrasted using 0.5% OsO$_4$ (Roth, Germany; RT, 1.5 hr) and 1% uranyl acetate (Polysciences, Germany) in 70% ethanol (RT, 1 hr). After dehydration tissue blocks were embedded in epoxy resin (Durcopan, Roth, Germany) and ultrathin sections of 40 nm thickness were cut using a Leica UC6 ultramicrotome (Leica, Wetzlar, Germany). Sections were imaged using a Zeiss 906 TEM (Zeiss, Oberkochen, Germany) and analysed using ITEM software (Olympus, Germany).

## $^1$H NMR spectroscopy-based metabonomic analysis

Polar metabolites were extracted from the gastrocnemius muscle using previously described protocols (*Beckonert et al., 2007*). Briefly, 40–50 mg of muscle tissue was snap frozen in liquid nitrogen and finely ground in 300 μL of chloroform: methanol (2:1) using a tissue lyzer. The homogenate was combined with 300 μL of water, vortexed and spun (13,000 g for 10 min) to separate the aqueous (upper) and organic (lower) phases. A vacuum concentrator (SpeedVac) was used to remove the water and methanol from the aqueous phase before reconstitution in 550 μL of phosphate buffer (pH 7.4) in 100% D$_2$O containing 1 mM of the internal standard, 3-(trimethylsilyl)-[2,2,3,3,$^{-2}$H$_4$]-propionic acid (TSP). For each sample, a standard one-dimensional NMR spectrum was acquired with water peak suppression using a standard pulse sequence (recycle delay (RD)-90°-$t_1$-90°-$t_m$-90°-acquire free induction decay (FID)). RD was set as 2 s, the 90° pulse length was 16.98 μs, and the mixing time ($t_m$) was 10 ms. For each spectrum, 8 dummy scans were followed by 128 scans with an acquisition time per scan of 3.8 s and collected in 64 K data points with a spectral width of 12.001 ppm. $^1$H nuclear magnetic resonance (NMR) spectra were manually corrected for phase and baseline distortions and referenced to the TSP singlet at δ 0.0. Spectra were digitized using an in-house MATLAB (version R2009b, The Mathworks, Inc.; Natwick, MA) script. To minimize baseline distortions arising from imperfect water saturation, the region containing the water resonance was excised from

the spectra. Principal components analysis (PCA) was performed with Pareto scaling in MATLAB using scripts provided by Korrigan Sciences Ltd, UK.

## Protein expression by western blotting

Frozen muscles were powdered and lysed in a buffer containing 50 mM Tris, pH7.5, 150 mM NaCl, 5 mM MgCl₂, 1 mM DTT, 10% glycerol, 1%SDS, 1%Triton X-100, 1XRoche Complete Protease Inhibitor Cocktail, 1X Sigma-Aldrich Phosphatase Inhibitor Cocktail 1 and 3. Then, the samples were immunoblotted and visualized with SuperSignal West Pico Chemiluminescent substrate (Pierce). Blots were stripped using Restore Western Blotting Stripping Buffer (Pierce) according to the manufacturer's instructions and were reprobed if necessary. Details of antibodies are given in *Supplementary file 1*.

## Quantitative PCR

Tissue samples were solubilised in TRIzol (Fisher) using a tissue homogeniser. Total RNA was prepared from skeletal muscles using the RNeasy Mini Kit (Quigen, Manchester, UK ). Total RNA (5 µg) was reverse-transcribed to cDNA with SuperScript II Reverse Transcriptse (Invitrogen) and analyzed by quantitative real-time RT-PCR on a StepOne Plus cycler, using the Applied Biosystems SYBR-Green PCR Master Mix. Primers were designed using the software Primer Express 3.0 (Applied Biosystems). Relative expression was calculated using the $\Delta\Delta C_t$ method with normalization to the housekeeping genes cyclophilin-B, hypoxanthine-guanine phosphoribosyltransferase (hprt) and glyceraldehyde-3-phosphate dehydrogenase (GAPDH). Specific primer sequences are given in *Supplementary file 1*.

## Myonuclear organisation

For visualizing myonuclei, fibres were mounted with ProLong Diamons Antifade Mountant with DAPI (Molecular Probes, P36962), and a confocal microscope (Olympus FluoView 1000, BX61W1, Olympus, Japan) was used to observe single muscle fibres. Pictures were taken in confocal planes, separated by z-axis steps varying between 0.4 and 2 µm according to the optical thickness and the desired Nyquist sampling frequency. Confocal microscope images used for mapping of Euclidean positions of myonuclei were processed and analysed using Imaris (Bitplane) and ImageJ (NIH, Bethesda, MD, USA).

For each muscle fibre, an idealized circular cylinder segment with constant radius was constructed, and the distance from each nucleus to its nearest neighbour was calculated.

In order to measure how ordered the nuclei distribution for a particular fibre is, the mean nearest neighbour distance was calculated for the experimental data, as well as for the random and optimal distribution using parameters from the experiment. We denote the experimental, random and optimal means by ME, MR and MO. An 'orderness-score', g(ME), was then calculated as:

$$g(M_E, M_R, M_O) = \frac{M_E - M_R}{M_O - M_R}$$

Further details and availability of custom made software, please contact j.c.bruusgaard@ibv.uio.no.

## Satellite cell culture

Single fibres from EDL were isolated using 0.2% collagenase I in DMEM medium and either fixed or cultured for 48 and 72 hr as previously described (*Otto et al., 2008*).

## Skeletal muscle regeneration

Skeletal muscle damage was induced by injecting 30 µl of 50 µM cardiotoxin in the tibialis anterior (TA) muscle of one limb while the contralateral TA of the other limb was injected with 30 µL PBS to serve as an internal control. The degree of muscle regeneration was assessed on day 3 and day 6 post-injury.

## Statistical analysis

Data are presented as mean ± SE. Significant differences between two groups were performed by Student's t-test for independent variables. The normality of the data for two samples was checked

with a Kolmogorov–Smirnov test ($\alpha$ = 10%). Differences among groups were analysed by one-way or two-way analysis of variance (ANOVA) followed by Bonferroni's multiple comparison tests as appropriate. In the case of non-homogeneous variances (Lavene's test; $p < 0.05$) for a variable, ANOVA was performed using the square root of the observations. Statistical analysis was performed on SPSS 18.0 (Chicago, IL). Differences were considered statistically significant at $p < 0.05$.

## Acknowledgements

The financial support from the Biotechnology and Biological Sciences Research Council is gratefully acknowledged (Grants BB/J016454/1 to HCH and BB/I015787/1 to RM). The study was also supported by the European Union and The Royal Society (Grants: FP7-PEOPLE-PCIG14-GA-2013-631440 and RG140470 Research Grant to AM).

## Additional information

### Funding

| Funder | Grant reference number | Author |
|---|---|---|
| European Commission | FP7-PEOPLE-676 | Antonios Matsakas |
| European Commission | PCIG14-GA-2013-631440 | Antonios Matsakas |
| Royal Society | Research Grant, RG140470 | Antonios Matsakas |
| Biotechnology and Biological Sciences Research Council | BB/I015787/1 | Robert Mitchell |
| Biotechnology and Biological Sciences Research Council | BB/J016454/1 | Henry Collins-Hooper |

The funders had no role in study design, data collection and interpretation, or the decision to submit the work for publication.

### Author contributions

SO, Acquisition of data, Analysis and interpretation of data; AM, NG, Acquisition of data, Analysis and interpretation of data, Drafting or revising the article; HD, Experimental design, Experimentation, data analysis, Manuscript preparation; OK, JCB, Experimentation, data analysis, Manuscript preparation; K-AH, AVS, Experimentation, data analysis; BJ, Experimentation; RS, RM, HC-H, Acquisition of data; KF, Analysis and interpretation of data, Drafting or revising the article; AP, OR, JRS, Conception and design; MS, Drafting or revising the article; VN, Conception and design, Drafting or revising the article; TBH, Conception and design, Analysis and interpretation of data, Drafting or revising the article; KP, Conception and design, Acquisition of data, Analysis and interpretation of data, Drafting or revising the article

### Author ORCIDs

Ketan Patel, http://orcid.org/0000-0002-7131-749X

### Ethics

Animal experimentation: The experiments were performed under a project license (PPL70/7516) from the United Kingdom Home Office in agreement with the Animals (Scientific Procedures) Act 1986. The University of Reading Animal Care and Ethical Review Committee approved all procedures. All of the animals were handled according to approved institutional animal care and guidelines set out by the Home Office of the UK. The protocol was approved by the Committee on the Ethics of Animal Experiments of the University of reading. All surgery was performed under recommended anesthesia, and every effort was made to minimize suffering.

## Additional files

**Supplementary files**
• Supplementary file 1. List of primary and secondary antibodies and qPCR primer sequences.

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
