## [Decision Letter]

Thank you for submitting your article "Enhanced exercise and regenerative capacity in a model that violates size constraints of oxidative muscle fibres" for consideration by *eLife*. Your article has been reviewed by three peer reviewers, one of whom is a member of our Board of reviewing Editors, and the evaluation has been overseen by K VijayRaghavan as the Senior Editor. The following individuals involved in review of your submission have agreed to reveal their identity: Gillian Butler-Browne (Reviewer #3).

The reviewers have discussed the reviews with one another and the Reviewing Editor has drafted this decision to help you prepare a revised submission.

All reviewers concur on the novelty and interest of your work that shows for the first time that a slow twitching oxidative phenotype can be independent from muscle fibre size. They also agree on the interest of other aspects of the work, such as the reduced number of satellite cells in the double mutant that however does not affect muscle regeneration.

However, two reviewers find that the manuscript is spoiled by poor proofreading, some careless statements that need to be corrected and some statistical assumptions that need to be addressed. They are listed below:

Essential revisions:

1) Subsection “Ultra-structure”, first paragraph. The authors state that "Depletion of mitochondria is often associated with compensatory hypertrophy". They should supply a reference to support this statement and rewrite the statement as it currently suggests that muscle hypertrophies in response to depletion of mitochondria which appears extremely unlikely.

2) Second paragraph of the same section. The authors state that "Over-expression of Errγ in the Mtn-/- reverses almost all the ultra-structural abnormalities." That is an incorrect expression. Over-expression of Errγ prevents development of almost all the ultra-structural abnormalities seen in the Mtn-/- mouse. There is a big difference between reversal and prevention as the former suggests that the abnormalities form first and are then reversed and the authors show no evidence to support this.

3) Subsection “Statistical analysis”. The statistical analyses appear to have assumed a normal distribution and used parametric tests without checking that the distribution was indeed normal or an approximation of normal. It is recommended that the authors go back and test this assumption for each data set. The authors should also report the statistical software used for the calculations.

4) Figure legends. One-way ANOVA, *P<0.05 is not an adequate statement as the ANOVA only reveals that there are differences between groups and NOT which groups are different. The correct statement is: "One-way ANOVA followed by Bonferroni's multiple comparison tests, *P<0.05". However this may need to be modified depending on whether the data are normally distributed (see point 3 above).

---

## [Author Response]

*Essential revisions:*

*1) Subsection “Ultra-structure”, first paragraph. The authors state that "Depletion of mitochondria is often associated with compensatory hypertrophy". They should supply a reference to support this statement and rewrite the statement as it currently suggests that muscle hypertrophies in response to depletion of mitochondria which appears extremely unlikely.*

We have clarified this section as follows:

Mitochondrial hypertrophy has been postulated to compensate for decreased mitochondrial number or function. Hypertrophy is thought to either to protect against apoptosis or for functional mitochondria to fuse with aberrant ones resulting in the maintenance of cell function (Frank et al., 2001, Ono et al., 2001). Mitochondrial hypertrophy was evident in both compartments in muscle from Mtn-/- (Figure 5) and was normalized by Errγ in the sub-membrane region (Figure 5).

and provided the following references:

Ono, Isobe, Nakada and Hayashi, 2001. Human cells are protected from mitochondrial dysfunction by complementation of DNA products in fused mitochondria. Nat Genet, 28, 272-5.

Frank et al. J. 2001. The role of dynamin-related protein 1, a mediator of mitochondrial fission, in apoptosis. Dev Cell, 1, 515-25.

*2) Second paragraph of the same section. The authors state that "Over-expression of Errγ in the Mtn-/- reverses almost all the ultra-structural abnormalities." That is an incorrect expression. Over-expression of Errγ prevents development of almost all the ultra-structural abnormalities seen in the Mtn-/- mouse. There is a big difference between reversal and prevention as the former suggests that the abnormalities form first and are then reversed and the authors show no evidence to support this.*

We have made the changes as requested the reviewers.

3) Subsection “Statistical analysis”. The statistical analyses appear to have assumed a normal distribution and used parametric tests without checking that the distribution was indeed normal or an approximation of normal. It is recommended that the authors go back and test this assumption for each data set. The authors should also report the statistical software used for the calculations.

We have indeed checked our data for normal distribution but mistakenly the relevant statement was not included in the initial submission. We have now revised this section as follow:

“Normality of the data for two samples was checked with a Kolmogorov–Smirnov test (α =10%). Differences among groups were analysed by one-way or two-way analysis of variance (ANOVA) followed by Bonferroni’s multiple comparison tests as appropriate. In the case of non-homogeneous variances (Lavene’s test; P <0.05) for a variable, ANOVA was performed using the square root of the observations. Statistical analysis was performed on SPSS 18.0 (Chicago, IL)”

4) Figure legends. One-way ANOVA, *P<0.05 is not an adequate statement as the ANOVA only reveals that there are differences between groups and NOT which groups are different. The correct statement is: "One-way ANOVA followed by Bonferroni's multiple comparison tests, *P<0.05". However this may need to be modified depending on whether the data are normally distributed (see point 3 above).

We thank the reviewer for this suggestion. We have revised all the figure legends as recommended to give much more information regarding the degree of statistical significance.